# BINARY HYPOTHESIS TESTING FOR SOFTMAX MODELS AND LEVERAGE SCORE MODELS

## ABSTRACT

Softmax distributions are widely used in machine learning, including Large Language Models (LLMs) where the attention unit uses softmax distributions. We abstract the attention unit as the softmax model, where given a vector input, the model produces an output drawn from the softmax distribution (which depends on the vector input). We consider the fundamental problem of binary hypothesis testing in the setting of softmax models. That is, given an unknown softmax model, which is known to be one of the two given softmax models, how many queries are needed to determine which one is the truth? We show that the sample complexity is asymptotically $O(\epsilon^{-2})$ where $\epsilon$ is a certain distance between the parameters of the models.

Furthermore, we draw analogy between the softmax model and the leverage score model, an important tool for algorithm design in linear algebra and graph theory. The leverage score model, on a high level, is a model which, given vector input, produces an output drawn from a distribution dependent on the input. We obtain similar results for the binary hypothesis testing problem for leverage score models.

## 1 INTRODUCTION

In transforming various aspects of people's lives, large language models (LLMs) have exhibited tremendous potential. In recent years, numerous content learning and LLMs have been developed, including notable models such as Adobe Firefly, Microsoft 365 Copilot (Spataro, 2023), Adobe Photoshop, and Google's Meena chatbot (Rathee, 2020), along with the GPT series and others (Radford et al., 2018; 2019; Devlin et al., 2018; Radford et al., 2019; Yang et al., 2019; Brown et al., 2020;?; ChatGPT, 2022; OpenAI, 2023). These models, together with those built upon them, have demonstrated significant prowess across diverse fields. The robustness and vitality of their development are attested to by the widespread integration of LLMs. In the realm of Natural Language Processing (NLP), evaluations by Liang et al. (2022); Laskar et al. (2023); Choi et al. (2023); Bang et al. (2023) center around natural language understanding, while Wang et al. (2023); Qin et al. (2023); Pu & Demberg (2023); Chia et al. (2023); Chen et al. (2023) delve into natural language generation. LLMs have found applications in diverse fields, including both social science and science (Guo et al., 2023; Deroy et al., 2023; Ferrara, 2023; Nay et al., 2023), medical applications (Chervenak et al., 2023; Johnson et al., 2023), and engineering (Pallagani et al., 2023; Sridhara et al., 2023; Bubeck et al., 2023; Liu et al., 2023b), showcasing their potent capabilities. A consistent theme among these models is the adoption of the transformer architecture, a proven and highly efficient framework. The prevailing prevalence of models like ChatGPT (OpenAI, 2023) further underscores the transformative impact of this architecture.

However, there is a crucial problem with LLMs: their training costs and uncertainty regarding their inference ability in different parts of the whole. Understanding how different domains work is important in retrieval argument generation (RAG) (Siriwardhana et al., 2023; Zamani & Bendersky, 2024; Salemi & Zamani, 2024), as well as sparsity for LLMs by identifying the ability domain in the model which is important in solving the problem above. Then a question arose:

*Can we distinguish different ability parts of large language models by limited parameters sampling?*

We take an initial step toward addressing this question from a theoretical perspective. As we delve deeper into LLMs, the softmax mechanism is found to play an important role in the computation of self-attention. Thus, it is imperative to study how the self-attention mechanism works, why it contributes significantly to the impressive capabilities of LLMs, and what role it plays are still not fully understood.

Therefore, in this work, we want to explore the mechanism of softmax distribution from a binary hypothesis testing perspective. By delving into the intricacies of the softmax formulation, we explore which parameters are important by explaining how the softmax can be distinguished from each other. By delving into this idea, we can determine how many parameters are important in the inference of transformers (Vaswani et al., 2017). In continuation of the paper and drawing upon a formulation similar to softmax, we also direct our attention to the distribution of leverage scores. Much like softmax, the leverage score is a distribution parameterized by a matrix. Both softmax and leverage score can be treated as functions of distribution within this context. Importantly, resembling softmax, leverage score assumes significance across various fields. Leverage scores have demonstrated their significant utility in both linear algebra and graph theory. In the field of graph theory, researchers have extensively explored the application of leverage scores in various areas such as the generation of random spanning trees (Schild, 2018), max-flow problems (Daitch & Spielman, 2008; Madry, 2013; 2016; Liu & Sidford, 2020), maximum matching (van den Brand et al., 2020a; Liu et al., 2020), and graph sparsification (Spielman & Srivastava, 2008a). Many studies have delved into the deep exploration of leverage scores, showcasing their effectiveness in optimization tasks such as linear programming (Lee & Sidford, 2014; van den Brand et al., 2020b), cutting-plane methods (Vaidya, 1989; Lee et al., 2015; Jiang et al., 2020b), semi-definite programming (Jiang et al., 2020a), and the approximation of the John Ellipsoid (Cohen et al., 2019). These applications underscore the importance of leverage scores in the context of theory of computer science and linear algebra. Based on the analysis provided, both the leverage score and softmax computation are parameterized by a single matrix. Given the significance of the application of softmax and computation, understanding the influence on parameter behavior becomes crucial. Hence, we delve into this inquiry by differentiating the model through parameter sampling and discussing how the number of samples affects the distinguishing ability.

A softmax model is parameterized by a matrix $A \in \mathbb{R}^{n \times d}$, and denoted $\texttt{SoftMax}_A$. Given $x \in \mathbb{R}^d$, the model outputs an element $i \in [n]$ with probability $p_i = \langle \exp(Ax), \mathbf{1}_n \rangle^{-1} \exp(Ax)_i$. In the binary hypothesis testing problem, we are given access to a softmax model which is either $\texttt{SoftMax}_A$ or $\texttt{SoftMax}_B$. We have query access to the model, that is, we can feed the model an input $x \in \mathbb{R}^d$, and it will produce an output. The goal is to determine whether the model is $\texttt{SoftMax}_A$ or $\texttt{SoftMax}_B$, using the fewest number of queries possible. We can similarly define the question for leverage score models. A leverage score model is parameterized by a matrix $A \in \mathbb{R}^{n \times d}$, and denoted $\texttt{Leverage}_A$. Given input $s \in (\mathbb{R} \backslash \{0\})^n$, the model returns an element $i \in [n]$ with probability $p_i = (A_s (A_s^\top A_s)^{-1} A_s^\top)_{i,i}/d$, where $A_s = S^{-1} A$, and $S = \text{Diag}(s)$ is the diagonal matrix with diagonal $s$. We define the binary hypothesis testing problem for leverage score models similarly to the softmax case.

## 1.1 MAIN RESULT.

We state informal versions of our main results.

**Theorem 1.1** (Informal statement of Theorem 3.2 and Theorem 3.5)**.** *Consider the binary hypothesis testing problem with two softmax models $\texttt{SoftMax}_A$ and $\texttt{SoftMax}_B$. We have 1). if $\|B - A\|_{2 \to \infty} \leq \epsilon$, then any successful algorithm uses $\Omega(\epsilon^{-2})$ queries (Lower bound), and 2). if $B = A + \epsilon M$ for some small $\epsilon$ then the hypothesis testing problem can be solved in $O(\epsilon^{-2} \nu)$ queries, where $\nu$ depends on $A$ and $M$ (Upper bound).*

**Theorem 1.2** (Informal statement of Theorem 4.2 and Theorem 4.3)**.** *Consider the binary hypothesis testing problem with two leverage score models $\texttt{Leverage}_A$ and $\texttt{Leverage}_B$. We have 1). if $\sum_{i \in [n]} \|B_{i,*}^\top B_{i,*} - A_{i,*}^\top A_{i,*}\|_{\text{op}} \leq \epsilon$, then any successful algorithm uses $\Omega(\epsilon^{-1})$ queries (Lower bound), and 2). if $B = A + \epsilon M$ for some small $\epsilon$ then the hypothesis testing problem can be solved in $O(\epsilon^{-2} \nu)$ queries, where $\nu$ depends on $A$ and $M$ (Upper bound).*

## 1.2 RELATED WORK

**Theoretical LLMs** Several investigations (Cai et al., 2021; Liu et al., 2023a; Reif et al., 2019; Hewitt & Manning, 2019) have concentrated on theoretical analyses concerning LLMs. The algorithm presented by Cai et al. (2021), named ZO-BCD, introduces a novel approach characterized by advantageous overall query complexity and reduced computational complexity in each iteration. The work by Liu et al. (2023a) introduces Sophia, a straightforward yet scalable second-order optimizer. Sophia demonstrates adaptability to curvature variations across different parameter regions, a feature particularly advantageous for language modeling tasks with strong heterogeneity. Importantly, the runtime bounds of Sophia are independent of the condition number of the loss function. Studies by Wang et al. (2022); Li & Liang (2021); Dai et al. (2021); Burns et al. (2022); Hase et al. (2023); Xie et al. (2022) investigate the knowledge and skills of LLMs. In the realm of optimization for LLMs, Kaplan et al. (2020); Cai et al. (2021); Rafailov et al. (2023); Liu et al. (2023a) have delved into this domain. Demonstrating the effectiveness of pre-trained models in localizing knowledge within their feed-forward layers, both Hase et al. (2023) and Meng et al. (2022) contribute valuable insights to the field. The exploration of distinct "skill" neurons and their significance in soft prompt-tuning for language models is a central theme in the analysis conducted by Wang et al. (2022), building upon the groundwork laid out in a prior discussion by Li & Liang (2021). The activation of skill neurons and their correlation with the expression of relevant facts is a focal point in the research presented by Dai et al. (2021), particularly in the context of BERT. In contrast, the work of Burns et al. (2022) takes an entirely unsupervised approach, leveraging the internal activations of a language model to extract latent knowledge. Lastly, the investigation by Li et al. (2022) sheds light on the sparsity observed in feedforward activations of large trained transformers, uncovering noteworthy patterns in their behavior. In addition to the above, Malladi et al. (2023); Deng et al. (2023a); Zelikman et al. (2023) explore Zero-th order algorithms for LLMs.

**Leverage Scores** Given $A \in \mathbb{R}^{n \times d}$ and $i \in [n]$, $a_i$ represents the $i$-th row of matrix $A$. We use $\sigma_i(A) = a_i^\top (A^\top A)^\dagger a_i$ to denote the leverage score for the $i$-th row of matrix $A$. The concept of leverage score finds extensive applications in the domains of machine learning and linear algebra. In numerical linear algebra and graph theory, leverage scores serve as fundamental tools. In the context of matrices, both the tensor CURT decomposition (Song et al., 2019) and the matrix CUR decomposition (Boutsidis & Woodruff, 2014; Song et al., 2017; 2019) heavily rely on leverage scores. In optimization, areas such as linear programming (Lee & Sidford, 2014; van den Brand et al., 2020b), the approximation of the John Ellipsoid (Cohen et al., 2019), cutting-plane methods (Vaidya, 1989; Lee et al., 2015; Jiang et al., 2020b), and semi-definite programming (Jiang et al., 2020a) incorporate leverage scores. Within graph theory applications, leverage scores play a crucial role in max-flow problems (Daitch & Spielman, 2008; Madry, 2013; 2016; Liu & Sidford, 2020), maximum matching (van den Brand et al., 2020a; Liu et al., 2020), graph sparsification (Spielman & Srivastava, 2008a), and the generation of random spanning trees (Schild, 2018). Several studies, such as Spielman & Srivastava (2008b); Drineas et al. (2012); Clarkson & Woodruff (2013), focus on the approximation of leverage scores. Simultaneously, Lewis weights, serving as a generalization of leverage scores, are explored in depth by Bourgain et al. (1989); Cohen & Peng (2015).

**Hypothesis Testing** Hypothesis testing is a central problem in statistics. In hypothesis testing, two (or more) hypotheses about the truth are given and an algorithm needs to distinguish which hypothesis is true. The most classic testing problem is the binary hypothesis testing. In this problem, two distributions $P_0$ and $P_1$ are given, and there is an unknown distribution $P$ which is either $P_0$ or $P_1$. The goal is to distinguish whether $P = P_0$ or $P = P_1$ by drawing samples from $P$. This problem is well-studied, with Neyman & Pearson (1933) giving tight characterization of the possible error regions in terms of the likelihood ratio. It is known that the asymptotic sample complexity of binary hypothesis testing for distributions is given by $\Theta(H^{-2}(P_0, P_1))$, where $H$ denotes the Hellinger distance, see e.g., Polyanskiy & Wu (2023+). There are other important kinds of hypothesis testing problems. In the goodness-of-fit testing problem, a distribution $Q$ is given, and there is an unknown distribution $P$ which is known to be either equal to $Q$ or far away from $Q$. The goal is to distinguish which is the true by drawing samples from $P$. In the two-sample testing problem, two unknown distributions $P$ and $Q$ are given, and it is known that either $P = Q$ or $P$ and $Q$ are far away from each other. The goal is to distinguish which is true by drawing samples from $P$ and $Q$. For these problems there are no simple general characterization as in the binary hypothesis testing. However, for reasonable classes of distributions such as Gaussian distributions or distributions on discrete

spaces, a lot of nice results are known (Ingster, 1987; 1982; Goldreich & Ron, 2011; Valiant & Valiant, 2017; Chan et al., 2014; Arias-Castro et al., 2018; Li & Yuan, 2019). We are not aware of any previous work that studies hypothesis testing problems for the class of softmax models or leverage score models.

**Roadmap.** In Section 2, we introduce notation and concepts related to information theory and hypothesis testing. Our results are presented in Section 3 and Section 4: Section 3 establishes upper and lower bounds on the sample complexity for distinguishing two different softmax models, and Section 4 delves into the case of leverage scores. We conclude and make further discussions in Section 5.

## 2 PRELIMINARIES

**Notation** Given $x \in \mathbb{R}^n$, we use $\|x\|_p$ to denote $\ell_p$ norm of $x$, where $\|x\|_0 = \sum_{i=1}^n \mathbb{1}(x_i \neq 0)$, $\|x\|_1 := \sum_{i=1}^n |x_i|$ ($\ell_1$ norm), $\|x\|_2 := (\sum_{i=1}^n x_i^2)^{1/2}$ ($\ell_2$ norm), and $\|x\|_\infty := \max_{i \in [n]} |x_i|$ ($\ell_\infty$ norm). For a square matrix, $\operatorname{tr}[A]$ is used to represent the trace of $A$. Given $1 \leq p \leq \infty$ and $1 \leq q \leq \infty$, $\|A\|_{p \to q}$ represents the $p$-to-$q$ operator norm $\|A\|_{p \to q} = \sup_{x:\|x\|_p \leq 1} \|Ax\|_q$. In particular, $\|A\|_{2 \to \infty} = \max_{i \in [n]} \|A_{i,*}\|_2$. For $x \in \mathbb{R}^n$, let $\operatorname{Diag}(x) \in \mathbb{R}^{n \times n}$ denote the diagonal matrix with diagonal $x$. For square matrix $A \in \mathbb{R}^{n \times n}$, let $\operatorname{diag}(A) \in \mathbb{R}^n$ denote the diagonal of $A$. For a non-negative integer $n$, let $[n]$ denote the set $\{1, \ldots, n\}$. For a sequence $X_1, \ldots, X_m$ of random variables, we use $X^m$ to denote the whole sequence $(X_1, \ldots, X_m)$.

### 2.1 INFORMATION THEORY

**Definition 2.1** (TV distance). *For two distributions $P, Q$ on the same measurable space, their total variation (TV) distance is $\operatorname{TV}(P, Q) = \frac{1}{2} \int |P(\mathrm{d}x) - Q(\mathrm{d}x)|$. In particular, if $P$ and $Q$ are on the discrete space $[n]$ and $P = (p_1, \ldots, p_n)$, $Q = (q_1, \ldots, q_n)$, then $\operatorname{TV}(P, Q)) = \frac{1}{2} \sum_{i=1}^n |p_i - q_i|$.*

**Definition 2.2** (Hellinger distance). *For two distributions $P, Q$ on the same measurable space, their squared Hellinger distance is $H^2(P, Q) = \frac{1}{2} \int (\sqrt{P(\mathrm{d}x)} - \sqrt{Q(\mathrm{d}x)})^2$. In particular, if $P$ and $Q$ are on the discrete space $[n]$ and $P = (p_1, \ldots, p_n)$, $Q = (q_1, \ldots, q_n)$, then*

$$H^2(P, Q) = \frac{1}{2} \sum_{i=1}^n (\sqrt{p_i} - \sqrt{q_i})^2 = 1 - \sum_{i=1}^n \sqrt{p_i q_i}.$$

*The Hellinger distance $H(P, Q)$ is the square root of the squared Hellinger distance $H^2(P, Q)$.*

We recall the following relationship between the Hellinger distance and the TV distance. For any distributions $P, Q$ on the same space, we have $H^2(P, Q) \leq \operatorname{TV}(P, Q) \leq \sqrt{2} H(P, Q)$.

**Definition 2.3** (Expectation and variance). *Let $P$ be a distribution on a measurable space $\mathcal{X}$ and $f$ be a continuous function on $\mathcal{X}$. Then $\mathbb{E}_P[f]$ is the expectation of $f$ under $P$ and $\operatorname{Var}_P(f)$ is the variance of $f$ under $P$. In particular, if $\mathcal{X} = [n]$, $P = (p_1, \ldots, p_n) \in \mathbb{R}^n$, and $x \in \mathbb{R}^n$, then $\mathbb{E}_P[x] = \sum_{i=1}^n p_i x_i$ and $\operatorname{Var}_P(x) = \sum_{i=1}^n p_i (x - \mathbb{E}_P[x])^2$.*

### 2.2 HYPOTHESIS TESTING

We review the classic hypothesis testing problem for distributions.

**Definition 2.4** (Binary hypothesis testing for distributions). *Let $P_0, P_1$ be two distributions on the same space. We have sample access to a distribution $P$, which is known to be either $P_0$ or $P_1$. The goal is to determine whether $P = P_0$ or $P = P_1$, using as few samples as possible. We say an algorithm successfully distinguishes $P_0$ and $P_1$ is at least $2/3$ under both hypotheses.*

In the above definition, the constant $2/3$ can be replaced by any constant $> 1/2$, and the asymptotic sample complexity of the binary hypothesis testing problem does not change. The reason is that if we have an algorithm that achieves success probability $\delta > \frac{1}{2}$, then we can run it independently a constant number of times and take the majority of the outputs. Thus, we can boost the success probability to an arbitrarily high constant. A classic result in information theory states that the sample complexity of the binary hypothesis testing problem is determined by the Hellinger distance.

**Lemma 2.5** (e.g., Polyanskiy & Wu (2023+)). *The sample complexity of the binary hypothesis testing problem for distributions is $\Theta(H^{-2}(P_0, P_1))$. That is, there is an algorithm that solves the problem using $O(H^{-2}(P_0, P_1))$ queries, and any algorithm that solves the problem uses $\Omega(H^{-2}(P_0, P_1))$ queries.*

## 2.3 SOFTMAX MODEL

**Definition 2.6** (Softmax model). *The softmax model $\mathtt{SoftMax}_A$ associated with $A \in \mathbb{R}^{n \times d}$ is a model such that on input $x \in \mathbb{R}^d$, it outputs a sample $y \in [n]$ from the distribution $\mathtt{SoftMax}_A(x)$, defined as follows: the probability mass of $i \in [n]$ is equal to $\langle \exp(Ax), \mathbf{1}_n \rangle^{-1} \exp(Ax)_i$.*

Note that $\sum_{i=1}^{n} \langle \exp(Ax), \mathbf{1}_n \rangle^{-1} \exp(Ax)_i = 1$, so the above definition gives a valid distribution.

**Definition 2.7** (Binary hypothesis testing for softmax models). *Let $A, B \in \mathbb{R}^{n \times d}$ be two matrices. Let $P_0 = \mathtt{SoftMax}_A, P_1 = \mathtt{SoftMax}_B$ be two softmax models. Let $P$ be the softmax model which is either $P_0$ or $P_1$. In each query, we can feed $x \in \mathbb{R}^d$ into $P$, and retrieve a sample $y \in [n]$ from $P(x)$. The goal is to determine whether the model $P$ is $P_0$ or $P_1$ in as few samples as possible. We say an algorithm successfully distinguishes $P_0$ and $P_1$, if the correctness probability is at least $2/3$ under both hypotheses.*

The above definition is valid. However, if we make no restrictions on the input $x$, then there would be undesirable consequences. For example, suppose $n = 2$, $d = 1$, $A = \begin{bmatrix} \epsilon \\ 0 \end{bmatrix}$, $B = \begin{bmatrix} 0 \\ \epsilon \end{bmatrix}$ for some very small $\epsilon > 0$. Because $A$ and $B$ are close to each other, we should expect it to be difficult to distinguish $\mathtt{SoftMax}_A$ and $\mathtt{SoftMax}_B$. However, if we allow any $x \in \mathbb{R}^d$ as input, then we could take $x$ to be a very large real number. Then $\mathtt{SoftMax}_A(x)$ has almost all mass on $1 \in [n]$, while $\mathtt{SoftMax}_B(x)$ has almost all mass on $2 \in [n]$, and we can distinguish the two models using only one query. To avoid this peculiarity, we assume that there is an energy constraint on $x$.

**Definition 2.8** (Energy constraint for softmax model). *We assume that there is an energy constraint, that is, input $x \in \mathbb{R}^n$ should satisfy $\|x\|_2 \leq E$, for some given constant $E$.*

The energy constraint is a reasonable assumption in the context of LLMs and more generally neural networks, because of the widely used batch normalization technique (Ioffe & Szegedy, 2015).

## 2.4 LEVERAGE SCORE MODEL

**Definition 2.9** (Leverage score model). *The leverage score model $\mathtt{Leverage}_A$ associated with $A \in \mathbb{R}^{n \times d}$ is a model such that on input $s \in (\mathbb{R} \setminus \{0\})^n$, it outputs a sample $y \in [n]$ from the distribution $\mathtt{Leverage}_A(s)$, defined as follows: the probability mass of $i \in [n]$ is equal to*

$$\|(A_s^\top A_s)^{-1/2}(A_s)_{*,i}\|_2^2 / d = (A_s(A_s^\top A_s)^{-1} A_s^\top)_{i,i} / d,$$

*where $A_s = S^{-1}A$, and $S = \mathrm{Diag}(s)$.*

**Definition 2.10** (Binary hypothesis testing for leverage score model). *Let $A, B \in \mathbb{R}^{n \times d}$ be two matrices. Let $P_0 = \mathtt{Leverage}_A, P_1 = \mathtt{Leverage}_B$ be two leverage score models. Let $P$ be the leverage score model which is either $P_0$ or $P_1$. In each query, we can feed $s \in (\mathbb{R} \setminus \{0\})^n$ into $P$, and retrieve a sample $y \in [n]$ from $P(s)$. The goal is to determine whether the model $P$ is $P_0$ or $P_1$ in as few samples as possible. We say an algorithm successfully distinguishes $P_0$ and $P_1$, if the correctness probability is at least $2/3$ under both hypotheses.*

Similar to the softmax model case, if we do not put any restrictions on $s$, then there will be certain weird behavior. For example, if we take $n = 2$, $d = 1$, $A = \begin{bmatrix} 1 \\ 0 \end{bmatrix}$ and $B = \begin{bmatrix} 1 \\ \epsilon \end{bmatrix}$ for some small $\epsilon > 0$. Because $A$ and $B$ are close to each other, we should expect it to be difficult to distinguish $\mathtt{Leverage}_A$ and $\mathtt{Leverage}_B$. However, if we allow any $s \in (\mathbb{R} \setminus \{0\})^n$ as input, then we can take $s = \begin{bmatrix} 1 & \delta \end{bmatrix}$ for some very small $\delta > 0$. In this way, we can verify that $\mathtt{Leverage}_A(s)$ has all mass on $1 \in [n]$, while $\mathtt{Leverage}_B(s)$ has almost all mass on $2 \in [n]$. So we can distinguish the two models using only one query. To avoid such cases we put additional constraints on $s$.

**Definition 2.11** (Constraint for leverage score model). *We assume that input $s \in (\mathbb{R} \setminus \{0\})^d$ should satisfy the constraint such that $c \leq s_i^2 \leq C$ for some given constants $0 < c < C$.*

## 3 SOFTMAX MODEL

### 3.1 GENERAL RESULT

We first prove a general result that relates the binary hypothesis testing problem with Hellinger distance, and the proof is deferred to Appendix A.1.

**Theorem 3.1.** *Let $A, B \in \mathbb{R}^{n \times d}$ be two matrices. Consider the binary hypothesis testing problem of distinguishing $\mathtt{SoftMax}_A$ and $\mathtt{SoftMax}_B$ using energy-constrained queries (Definition 2.8). Define $\delta = \sup_{x:\|x\|_2 \leq E} H(\mathtt{SoftMax}_A(x), \mathtt{SoftMax}_B(x))$. Then the sample complexity of the binary hypothesis testing problem is $\Theta(\delta^{-2})$. That is, there is an algorithm that successfully solves the problem using $O(\delta^{-2})$ energy-constrained queries, and any algorithm that successfully solves the problem uses $\Omega(\delta^{-2})$ energy-constrained queries.*

### 3.2 LOWER BOUND

Now, we prove the following lower bound for binary hypothesis testing for softmax models.

**Theorem 3.2** (Lower bound)**.** *If two softmax models (Definition 2.6) with parameters $A \in \mathbb{R}^{n \times d}$ and $B \in \mathbb{R}^{n \times d}$ satisfy $\|A - B\|_{2 \to \infty} \leq \epsilon$ (i.e., $\max_{j \in [n]} \|A_{j,*} - B_{j,*}\|_2 \leq \epsilon$), then any algorithm with energy constraint $E$ that distinguishes the two models with success probability $\geq \frac{2}{3}$ uses at least $\Omega(\epsilon^{-2} E^{-2})$ samples.*

Before giving the proof of Theorem 3.2, we state a lemma, and the proof is deferred to Appendix A.2.

**Lemma 3.3.** *Let $a, b \in \mathbb{R}^n$ be such that $\|a - b\|_\infty \leq \epsilon$. Let $P$ be the distribution on $[n]$ with $p_i = \exp(a_i)/\langle \exp(a), \mathbf{1}_n \rangle$. Let $Q$ be the distribution on $[n]$ with $q_i = \exp(b_i)/\langle \exp(b), \mathbf{1}_n \rangle$. Then*

$$H^2(P, Q) = O(\epsilon^2) \qquad \mathrm{TV}(P, Q) = O(\epsilon).$$

**Corollary 3.4.** *If matrices $A \in \mathbb{R}^{n \times d}, B \in \mathbb{R}^{n \times d}$ satisfy $\max_{j \in [n]} \|A_{j,*} - B_{j,*}\|_2 \leq \epsilon$, then for any $x \in \mathbb{R}^d$, the distributions $P = \mathtt{SoftMax}_A(x)$ and $Q = \mathtt{SoftMax}_B(x)$ satisfy*

$$H^2(P, Q) = O(\epsilon^2 \|x\|_2^2), \qquad \mathrm{TV}(P, Q) = O(\epsilon \|x\|_2).$$

*Proof.* For any $x \in \mathbb{R}^n$, we have

$$\|Ax - Bx\|_\infty = \max_{j \in [n]} |A_{j,*}x - B_{j,*}x| \leq \max_{h \in [n]} \|A_{j,*} - B_{j,*}\|_2 \|x\|_2 \leq \epsilon \|x\|_2.$$

The result then follows from Lemma 3.3. $\qquad \square$

*Proof of Theorem 3.2.* By Corollary 3.4, we have $H^2(\mathtt{SoftMax}_A(x), \mathtt{SoftMax}_B(x)) = O(\epsilon^2 E^2)$ for any $\|x\|_2 \leq E$. Therefore $\delta$ in the statement of Theorem 3.1 satisfies $\delta^2 = O(\epsilon^2 E^2)$. Applying Theorem 3.1 we finish the proof. $\qquad \square$

### 3.3 UPPER BOUND

In the previous section, we established an $\Omega(\epsilon^{-2})$ lower bound for solving the hypothesis testing problem for the softmax model. The upper bound is more subtle. Let us discuss a few difficulties in establishing the upper bound. Let $A, B \in \mathbb{R}^{n \times d}$ be parameters of the softmax models, $x \in \mathbb{R}^d$ be the input vector, $P = \mathtt{SoftMax}_A(x) = (p_1, \ldots, p_n)$, $Q = \mathtt{SoftMax}_B(x) = (q_1, \ldots, q_n)$. First, two different matrices $A$ and $B$ could give rise to the same softmax model. If $B = A + \mathbf{1}_n^\top w$ for some $w \in \mathbb{R}^d$, then for any $x \in \mathbb{R}^d$, we have

$$q_i = \frac{\exp(Bx)_i}{\langle \exp(Bx), \mathbf{1}_n \rangle} = \frac{\exp(Ax)_i \exp(w^\top x)}{\langle \exp(Ax) \exp(w^\top x), \mathbf{1}_n \rangle} = \frac{\exp(Ax)_i}{\langle \exp(Ax), \mathbf{1}_n \rangle} = p_i$$

for all $i \in [d]$. Therefore in this case $\mathtt{SoftMax}_A(x) = \mathtt{SoftMax}_B(x)$ for all $x \in \mathbb{R}^d$ and it is impossible to distinguish the two models. This issue may be resolved by adding additional assumptions such as $\mathbf{1}_n^\top A = \mathbf{1}_n^\top B$. A more important issue is that $A$ and $B$ may differ only in rows with very small probability weight under any input $x$. For example, suppose $A$ is the zero matrix, and $B$

differ with $A$ only in the first row. For any $x \in \mathbb{R}^d$, the distribution $\texttt{SoftMax}_A(x)$ is the uniform distribution on $[d]$. If $\|B_{1,*} - A_{1,*}\|_2 = \epsilon$, then for any $x$ with $\|x\|_2 \leq E$, we have

$$\exp(-\epsilon E) \leq \frac{\exp(Bx)_1}{\exp(Ax)_1} \leq \exp(\epsilon E).$$

A simple calculation shows that in this case, $H^2(P, Q) = O(\epsilon^2 E^2/n)$. So the sample complexity of any hypothesis testing algorithm is at least $\Omega(n/(\epsilon^2 E^2))$, which grows with $n$. This shows that the sample complexity may depend on $n$. Nevertheless, using Theorem 3.1, we show a local upper bound, which says that for fixed $A$ and fixed direction $M$, there is an algorithm that distinguishes $\texttt{SoftMax}_A$ and $\texttt{SoftMax}_{A+\epsilon M}$ using $O(\epsilon^{-2})$ queries, for small enough $\epsilon > 0$.

**Theorem 3.5.** *Fix $A, M \in \mathbb{R}^{n \times d}$ where $\|M\|_{2 \to \infty} = O(1)$. For $\epsilon > 0$, define $B_\epsilon = A + \epsilon M$. We consider the binary hypothesis testing problem with $\texttt{SoftMax}_A$ and $\texttt{SoftMax}_{B_\epsilon}$, for small $\epsilon$. Let $\nu = \sup_{x:\|x\|_2 \leq E} \text{Var}_{\texttt{SoftMax}_A(x)}(Mx)$. Then for $\epsilon > 0$ small enough, there is an algorithm that uses $O(\epsilon^{-2}\nu^{-1})$ energy-constrained queries and distinguishes between $\texttt{SoftMax}_A$ and $\texttt{SoftMax}_{B_\epsilon}$.*

Proof of Theorem 3.5 is deferred to Appendix A.3. From Theorem 3.5 we see that it is an interesting problem to bound $\nu = \sup_{x:\|x\|_2 \leq E} \text{Var}_{\texttt{SoftMax}_A(x)}(Mx)$ for fixed $A, M \in \mathbb{R}^{n \times d}$. For different $A$ and $M$ the value of $\nu$ can be quite different. For example, if $A$ is the all zero matrix and $M$ is zero except for row 1 (and $\|M\|_{2 \to \infty} = O(1)$), then $\nu = O(E^2/n)$ for any $\|x\|_2 \leq E$. On the other hand, if $A$ is the zero matrix, and the first column $M$ are i.i.d. Gaussian $\mathcal{N}(0, \Theta(1))$, then with high probability, $\nu = \Omega(E^2)$ for $x = (E, 0, \ldots, 0)$. We remark that Theorem 3.5 is in fact tight. We have a matching lower bound.

**Theorem 3.6.** *Under the same setting as Theorem 3.5, for sufficient small $\epsilon > 0$, any algorithm that distinguishes between $\texttt{SoftMax}_A$ and $\texttt{SoftMax}_{B_\epsilon}$ must use $\Omega(\epsilon^{-2}\nu^{-1})$ energy-constrained queries.*

*Proof.* It follows from combining the proof of Theorem 3.5 and Theorem 3.1. $\square$

## 4 LEVERAGE SCORE MODEL

### 4.1 GENERAL RESULT

We first prove a general result which is the leverage score version of Theorem 3.1.

**Theorem 4.1.** *Let $A, B \in \mathbb{R}^{n \times d}$ be two matrices. Consider the binary hypothesis testing problem of distinguishing $\texttt{Leverage}_A$ and $\texttt{Leverage}_B$ using constrained queries (Definition 2.11). Define $\delta = \sup_{s:c \leq s_i^2 \leq C \forall i} H(\texttt{Leverage}_A(s), \texttt{Leverage}_B(s))$. Then the sample complexity of the binary hypothesis testing problem is $\Theta(\delta^{-2})$. That is, there is an algorithm that successfully solves the problem using $O(\delta^{-2})$ energy-constrained queries, and any algorithm that successfully solves the problem uses $\Omega(\delta^{-2})$ energy-constrained queries.*

*Proof.* The proof is similar to Theorem 3.1 and omitted. $\square$

### 4.2 LOWER BOUND

The goal of this section is to prove the following lower bound for binary hypothesis testing for leverage score models.

**Theorem 4.2.** *Consider two leverage score model $\texttt{Leverage}_A$ and $\texttt{Leverage}_B$. Assume that there exists $\delta > 0$ such that $A^\top A \succeq \delta I$. If $\sum_{i \in [n]} \|B_{i,*}^\top B_{i,*} - A_{i,*}^\top A_{i,*}\|_{\text{op}} \leq \epsilon$ (where $\|\cdot\|_{\text{op}}$ denotes the 2-to-2 operator norm), then any algorithm that solves the binary hypothesis testing problem takes at least $\Omega(c\delta/(C\epsilon))$ constrained queries.*

*Proof.* Let $P = \texttt{Leverage}_A(s) = (p_1, \ldots, p_n)$ and $Q = \texttt{Leverage}_B(s) = (q_1, \ldots, q_n)$. By Theorem 4.1, it suffices to prove that $H^2(P, Q) = O(\epsilon C/(c\delta))$. We first consider the case where $A$ and $B$ differ in exactly one row $i$. Fix $s \in \mathbb{R}^d$ with $c \leq s_j \leq C$ for all $j \in [n]$. Let $A_s = S^{-1}A$ and $B_s = S^{-1}B$, where $S = \text{Diag}(s)$.

Because $A^\top A \succeq \delta I$, we have $A_s^\top A_s \succeq (\delta/C) \cdot I$. Because $\|B_{i,*}^\top B_{i,*} - A_{i,*}^\top A_{i,*}\|_{\mathrm{op}} \leq \epsilon$, we have

$$-\epsilon_i C/\delta A_s^\top A_s \preceq B_{i,*}^\top B_{i,*} - A_{i,*}^\top A_{i,*} \preceq \epsilon_i C/\delta A_s^\top A_s.$$

Recall that $A$ and $B$ differ in exactly one row $i$. Therefore

$$(1 - \frac{\epsilon C}{c\delta})A_s^\top A_s \preceq B_s^\top B_s \preceq (1 + \frac{\epsilon C}{c\delta})A_s^\top A_s. \tag{1}$$

For $j \neq i$, we have

$$\begin{aligned}
q_j &= s_j^{-2} B_{j,*}(B_s^\top B_s)^{-1}(B^\top)_{*,j}/d \\
&= \mathrm{tr}[s_j^{-2}(B^\top)_{*,j}B_{j,*}(B_s^\top B_s)^{-1}]/d \\
&= (1 \pm O(\epsilon C/(c\delta)))\,\mathrm{tr}[s_j^{-2}A_{j,*}^\top A_{j,*}(A_s^\top A_s)^{-1}]/d \\
&= (1 \pm O(\epsilon C/(c\delta)))p_j, \tag{2}
\end{aligned}$$

where the first step is by definition of the leverage score model, the second step is by property of trace, the third step is Eq. (1), the fourth step is by definition of the leverage score model.

**Upper bound for** TV. For the TV distance, we have

$$\mathrm{TV}(P,Q) = \frac{1}{2}\sum_{j=1}^n |p_j - q_j| \leq \sum_{j \neq i} |p_j - q_j| \leq \sum_{j \neq i} O(\epsilon C/(c\delta))p_i \leq O(\epsilon C/(c\delta)).$$

where the first step is by definition of TV distance, the third step is by Eq. (2). Therefore $\mathrm{TV}(P,Q) \leq O(\epsilon C/(c\delta))$.

**Upper bound for** $H^2(P,Q)$. Using $H^2(P,Q) \leq \mathrm{TV}(P,Q)$ we also get $H^2(P,Q) \leq O(\epsilon C/(c\delta))$.

Now we have established the result when $A$ and $B$ differ in exactly one row. Let us now consider general case. If $\epsilon \geq 0.1\delta$, then $c\delta/(C\epsilon) = O(1)$ and there is nothing to prove. In the following, assume that $\epsilon \leq 0.1\delta$. For $0 \leq k \leq n$, define $B^k \in \mathbb{R}^{n \times d}$ be the matrix with $B_{i,*}^k = B_{i,*}$ for $i \leq k$ and $B_{i,*}^k = A_{i,*}$ for $i \geq k$. Then $B^0 = A$, $B^n = B$, and $B^k$ and $B^{k+1}$ differ exactly in one row. Let $\epsilon_i = \|B_{i,*}^\top B_{i,*} - A_{i,*}^\top A_{i,*}\|_{\mathrm{op}}$. Then by the above discussion, we have

$$\mathrm{TV}(\texttt{Leverage}_{B^k}(s), \texttt{Leverage}_{B^{k+1}}(s)) = O(\epsilon_k C/(c\delta))$$

for all $0 \leq k \leq n-1$. By metric property of TV, we have

$$\begin{aligned}
\mathrm{TV}(P,Q) &\leq \sum_{0 \leq k \leq n-1} \mathrm{TV}(\texttt{Leverage}_{B^k}(s), \texttt{Leverage}_{B^{k+1}}(s)) \\
&= \sum_{0 \leq k \leq n-1} O(\epsilon_i C/(c\delta)) \\
&= O(\epsilon C/(c\delta)).
\end{aligned}$$

Using $H^2(P,Q) \leq \mathrm{TV}(P,Q)$ we also get $H^2(P,Q) = O(\epsilon C/(c\delta))$. This finishes the proof. $\qquad \square$

In Theorem 4.2, the bound has linear dependence in $\epsilon^{-1}$. An interesting question is the improve the bound to quadratic dependence $\epsilon^{-2}$.

### 4.3 Upper Bound

Let $A, B \in \mathbb{R}^{n \times d}$ be parameters of the leverage score models, $s \in \mathbb{R}^n$ be the input vector, $P = \texttt{Leverage}_A(s) = (p_1, \ldots, p_n)$, $Q = \texttt{Leverage}_B(s) = (q_1, \ldots, q_n)$. For the upper bounds of the leverage score model, we run into similar difficulties as for the softmax model. Firstly, different matrices $A$ and $B$ could give rise to the same leverage score model. If $B = AR$ for some invertible matrix $R \in \mathbb{R}^{d \times d}$, then we have

$$q_i = (B_s(B_s^\top B_s)^{-1}B_s^\top)_{i,i}/d = (A_s R(R^\top A_s^\top A_s R)^{-1}R^\top A_s^\top)_{i,i}/d = (A_s(A_s^\top A_s)^{-1}A_s^\top)_{i,i}/d = p_i.$$

Then $\texttt{Leverage}_A(s) = \texttt{Leverage}_B(s)$ for all $s \in (\mathbb{R}\backslash\{0\})^n$ and it is impossible to distinguish the two models. Furthermore, there exist scenarios where $A$ and $B$ differ only in rows with very small

probability weight under any input $s$. We now give an example where $\|A_{1,*}^\top A_{1,*} - B_{1,*}^\top B_{1,*}\| = \Omega(1)$ but $\mathrm{TV}(\texttt{Leverage}_A(s), \texttt{Leverage}_B(s)) = O(1/n)$ for any $s$ satisfying $c \leq s_i^2 \leq C$ for all $i \in [n]$. Suppose $A = [I_d \quad e_1 \quad \cdots \quad e_1]^\top$ (that is, the first $d$ rows of $A$ is equal to $I_d$, and all remaining rows are equal to $e_1^\top = (1, 0, \ldots, 0)$). Then for $s$ satisfying $c \leq s_i^2 \leq C$ for all $i \in [n]$, the distribution $P = \texttt{Leverage}_A(s)$ has probability mass $O(1/n)$ on every element $i \in \{1, d+1, d+2 \ldots, n\}$ (hiding constants depending on $c$ and $C$). Now suppose $B$ differs with $A$ only in the first entry $(1, 1)$, and $B_{1,1} = A_{1,1} + \Theta(1)$. Then for fixed $s$, $q_j = p_j$ for $j \in \{2, \ldots, d\}$, $q_1 \geq p_1$, and $q_j \leq p_j$ for $j \in \{d+1, \ldots, n\}$. So $H^2(P, Q) \leq \mathrm{TV}(P, Q) = q_1 - p_1 = \Theta(1/n)$. This shows that the sample complexity may depend on $n$. After discussing the difficulties in establishing an upper bound, we now show a local upper bound, which says for fixed $A$ and fixed direction $M$, there is an algorithm that distinguishes $\texttt{Leverage}_A$ and $\texttt{Leverage}_{A+\epsilon M}$ using $O(\epsilon^{-2})$ queries, for small enough $\epsilon > 0$.

**Theorem 4.3.** *Fix $A, M \in \mathbb{R}^{n \times d}$ where $\|M\|_{2 \to \infty} = O(1)$. For $\epsilon > 0$, define $B_\epsilon = A + \epsilon M$. We consider the binary hypothesis testing problem with $\texttt{Leverage}_A$ and $\texttt{Leverage}_{B_\epsilon}$, for small $\epsilon$. Let $\nu = \sup_s \mathrm{Var}_{\texttt{Leverage}_A(s)}(w_s)$ where*

$$w_s = \frac{\mathrm{diag}((I - A_s(A_s^\top A_s)^{-1} A_s^\top)(M_s(A_s^\top A_s)^{-1} A_s^\top))}{\mathrm{diag}(A_s(A_s^\top A_s)^{-1} A_s^\top)}$$

*where the division between vectors is entrywise division. Then for $\epsilon > 0$ small enough, there is an algorithm that uses $O(\epsilon^{-2} \nu^{-1})$ queries and distinguishes between $\texttt{Leverage}_A$ and $\texttt{Leverage}_{B_\epsilon}$.*

Proof of Theorem 4.3 is deferred to Appendix A.4. Similarly to the softmax model case, Theorem 4.3 is also tight.

**Theorem 4.4.** *Work under the same setting as Theorem 4.3. For $\epsilon > 0$ small enough, any algorithm that distinguishes between $\texttt{SoftMax}_A$ and $\texttt{SoftMax}_{B_\epsilon}$ must use $\Omega(\epsilon^{-2} \nu^{-1})$ energy-constrained queries.*

*Proof.* The proof is by combining the proof of Theorem 4.3 and Theorem 4.1. We omit the details. $\square$

## 5 CONCLUSION AND FUTURE DIRECTIONS

Widely applied across various domains, softmax and leverage scores play crucial roles in machine learning and linear algebra. This study delves into the testing problem aimed at distinguishing between different models of softmax and leverage score distributions, each parameterized by distinct matrices. We establish bounds on the number of samples within the defined testing problem. With the rapidly escalating computational costs in current machine learning research, our work holds the potential to offer valuable insights and guidance for distinguishing between the distributions of different models. We discuss a few possible directions for further research. In Theorem 3.5 and Theorem 4.3, we determine the local sample complexity of the binary hypothesis testing problems for softmax models and leverage score models. In particular, the sample complexity is $\Theta(\epsilon^{-2}\nu)$, where $\nu$ is a certain function depending on $A$ and $M$ (where $B = A + \epsilon M$). The form of $\nu$ is an optimization problem over the space of possible inputs. An interesting question is to provide bounds on the quantity $\nu$, or to provide computation-efficient algorithms for determining the value of $\nu$ of finding the optimal input ($x$ for softmax, $s$ for leverage score). This will lead to computation-efficient algorithms for solving the binary hypothesis testing problem in practice.

In this paper, we focused on the binary hypothesis testing problem, where the goal is to distinguish two models with different parameters. There are other hypothesis testing problems that are of interest both in theory and practice. For example, in the goodness-of-fit problem, the goal is to determine whether an unknown model is equal to or far away from a given model. In the two-sample testing problem, the goal is to determine whether two unknown models are the same or far away from each other. These problems have potential practical applications and we leave them as an interesting future direction.

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

# APPENDIX

# A MISSING PROOFS

## A.1 GENERAL RESULT FOR SOFTMAX MODEL

*Proof of Theorem 3.1.* **Lower bound.** If $\delta \geq 0.1$ then there is nothing to prove. In the following assume that $\delta < 0.1$. Suppose that there is an algorithm that successfully solves the binary hypothesis testing problem. Suppose it makes queries $x_1, \ldots, x_m \in \mathbb{R}^d$ where $x_i$ may depend on previous query results. Let $Y_1, \ldots, Y_m \in [n]$ denote the query results. Let $P_{Y^m}$ and $Q_{Y^m}$ denote the distribution of $Y^m$ under $P$ and $Q$, respectively. By definition of $\delta$, we have

$$H^2(P_{Y_k|Y^{k-1}}, Q_{Y_k|Y^{k-1}}) \leq \delta^2.$$

for any $k \in [m]$ and $Y^{k-1}$. Then

$$1 - H^2(P_{Y^m}, Q_{Y^m})$$
$$= \int \sqrt{P_{y^m} Q_{y^m}} \mathrm{d}y^m$$
$$= \int \sqrt{P_{y^{m-1}} Q_{y^{m-1}}}$$
$$\left( \int \sqrt{P_{y_m|y^{m-1}} Q_{y_m|y^{m-1}}} dy_m \right) \mathrm{d}y^{m-1}$$
$$\geq \int \sqrt{P_{y^{m-1}} Q_{y^{m-1}}} (1 - \delta^2) \mathrm{d}y^{m-1}.$$

Repeating this computation, in the end we get

$$1 - H^2(P_{Y^m}, Q_{Y^m}) \geq (1 - \delta^2)^m.$$

Because $\delta \leq 0.1$, we have $1 - \delta^2 \geq \exp(-2\delta^2)$. If $m \leq 0.01\delta^{-2}$, then

$$1 - H^2(P_{Y^m}, Q_{Y^m}) \geq \exp(-2\delta^2 m)$$
$$\geq \exp(-0.02) > 0.98,$$

and

$$H^2(P_{Y^m}, Q_{Y^m}) \leq 0.02.$$

This implies

$$\mathrm{TV}(P_{Y^m}, Q_{Y^m}) \leq \sqrt{2} H(P_{Y^m}, Q_{Y^m}) \leq 0.2,$$

which implies the success rate for binary hypothesis testing cannot be $\geq \frac{2}{3}$.

In conclusion, any algorithm that successfully solves the hypothesis testing problem need to use $\Omega(\delta^{-2})$ queries.

**Upper bound.** Take $x \in \mathbb{R}^d$ such that $\|x\|_2 \leq E$ and $\delta = H(\texttt{SoftMax}_A(x), \texttt{SoftMax}_B(x))$. By Lemma 2.5, using $O(\delta^{-2})$ samples we can distinguish $\texttt{SoftMax}_A(x)$ and $\texttt{SoftMax}_B(x)$. Therefore we can distinguish $\texttt{SoftMax}_A$ and $\texttt{SoftMax}_B$ in $O(\delta^{-2})$ queries by repeatedly querying $x$. □

## A.2 LOWER BOUND FOR SOFTMAX MODEL

Before giving the proof of Lemma 3.3, we prove a weaker version of the lemma.

**Lemma A.1.** *Let $a, b \in \mathbb{R}^n$. Suppose there exists an $\epsilon \geq 0$ such that for every $i \in [n]$, $b_i - a_i \in \{0, \epsilon\}$. Let $P$ be the distribution on $[n]$ with $p_i = \exp(a_i)/\langle \exp(a), \mathbf{1}_n \rangle$. Let $Q$ be the distribution on $[n]$ with $q_i = \exp(b_i)/\langle \exp(b), \mathbf{1}_n \rangle$. Then*

$$H^2(P, Q) = \frac{(1 - \exp(\epsilon/4))^2}{1 + \exp(\epsilon/2)} = O(\epsilon^2),$$
$$\mathrm{TV}(P, Q) = \tanh(\epsilon/4) = O(\epsilon).$$

*Proof.* Assume that $a$ and $b$ differ in $m$ coordinates. By permuting the coordinates, WLOG assume that $b_i = a_i + \epsilon$ for $1 \le i \le m$ and $b_i = a_i$ for $m + 1 \le i \le n$.

Write

$$s = \sum_{i=1}^{m} \exp(a_i)$$

and

$$t = \sum_{i=m+1}^{n} \exp(a_i).$$

Then

$$H^2(P, Q) = 1 - \sum_{i \in [n]} \sqrt{p_i q_i}$$

$$= 1 - \frac{s \exp(\epsilon/2) + t}{\sqrt{(s+t)(s \exp(\epsilon) + t)}}.$$

For fixed $t$ and $\epsilon$, the above is maximized at

$$s = t \exp(-\epsilon/2).$$

Plugging in the above $s$, we get

$$H^2(P, Q) \le 1 - \frac{2}{\sqrt{(\exp(-\epsilon/2) + 1)(\exp(\epsilon/2) + 1)}}$$

$$= \frac{(1 - \exp(\epsilon/4))^2}{1 + \exp(\epsilon/2)}.$$

For TV, we have

$$\mathrm{TV}(P, Q) = \sum_{m+1 \le i \le n} (q_i - p_i)$$

$$= \frac{t}{s+t} - \frac{t}{s \exp(\epsilon) + t}.$$

For fixed $t$ and $\epsilon$ the above is maximized at $s = t \exp(-\epsilon/2)$. Plugging in this $s$, we get

$$\mathrm{TV}(P, Q) \le \tanh(\epsilon/4).$$

$\square$

*Proof of Lemma 3.3.* We first prove the case where $b_i \ge a_i$ for all $i \in [n]$. Define $\epsilon_i = b_i - a_i$ for all $i \in [n]$. By permuting the coordinates, WLOG assume that $\epsilon_1 \le \cdots \le \epsilon_n$. Specially, define $\epsilon_0 = 0$. For $0 \le k \le n$, let $b^k \in \mathbb{R}^n$ denote the vector where $b_i^k = a_i + \min\{\epsilon_i, \epsilon_k\}$ for all $i \in [k]$. Then we can see that $b^0 = a$ and $b^n = b$, and for every $0 \le k \le n - 1$, the pair $(b^k, b^{k+1})$ satisfies the assumption in Lemma A.1. For $0 \le k \le n$, let $P^k$ denote the softmax distribution corresponding to $b^k$. By Lemma A.1, for every $0 \le k \le n - 1$, we have

$$H(P^k, P^{k+1}) = O(\epsilon_{k+1} - \epsilon_k),$$
$$\mathrm{TV}(P^k, P^{k+1}) = O(\epsilon_{k+1} - \epsilon_k).$$

Because Hellinger distance and TV distance are both metrics, we have

$$H(P, Q) = H(P^0, P^n)$$

$$\le \sum_{k=0}^{n-1} H(P^k, P^{k+1})$$

$$= O(\epsilon),$$

and

$$\mathrm{TV}(P, Q) = \mathrm{TV}(P^0, P^n)$$

$$\leq \sum_{k=0}^{n-1} \mathrm{TV}(P^k, P^{k+1})$$

$$= O(\epsilon).$$

This finishes the proof of the result when $b_i \geq a_i$ for all $i \in [n]$.

Now let us consider the general case. Let $c \in \mathbb{R}^n$ be defined as $c_i = \max\{a_i, b_i\}$ for all $i \in [n]$. Then

$$\max\{\|a - c\|_\infty, \|c - b\|_\infty\} \leq \|a - b\|_\infty \leq \epsilon.$$

Let $R$ be the softmax distribution corresponding to $c$. By our previous discussion, we have

$$H(P, R), H(R, Q), \mathrm{TV}(P, R), \mathrm{TV}(R, Q) = O(\epsilon).$$

By metric property of Hellinger distance and TV distance, we get

$$H(P, Q), H(P, Q) = O(\epsilon)$$

as desired.

$\square$

### A.3 LOCAL UPPER BOUND FOR SOFTMAX MODEL

*Proof of Theorem 3.5.* We take an $x$ satisfying $\|x\|_2 \leq E$ that maximizes $\mathrm{Var}_{\texttt{SoftMax}_A(x)}(Mx)$ and repeatedly query $x$. We would like to apply Theorem 3.1. To do that, we need to show that

$$H^2(\texttt{SoftMax}_A(x), \texttt{SoftMax}_{B_\epsilon}(x)) = \Omega(\epsilon^2 \nu).$$

Let $P = \texttt{SoftMax}_A(x) = (p_1, \ldots, p_n)$, $Q_\epsilon = \texttt{SoftMax}_{B_\epsilon}(x) = (q_{\epsilon,1}, \ldots, q_{\epsilon,n})$. Write $Z_A = \langle \exp(Ax), \mathbf{1}_n \rangle$, $Z_{B_\epsilon} = \langle \exp(B_\epsilon x), \mathbf{1}_n \rangle$.

Then, it follows that

$$Z_B = \sum_{j \in [n]} \exp(Ax)_j \exp(\epsilon(Mx)_j)$$

$$= \sum_{j \in [n]} \exp(Ax)_j + \sum_{j \in [n]} \exp(Ax)_j (\exp(\epsilon(Mx)_j) - 1)$$

$$= \sum_{j \in [n]} \exp(Ax)_j + \sum_{j \in [n]} \exp(Ax)_j (\epsilon(Mx)_j + O(\epsilon^2))$$

$$= Z_A (1 + \epsilon \langle p, Mx \rangle + O(\epsilon^2)). \tag{3}$$

where the initial step is because of $B = A + \epsilon M$, the second step is a result of simple algebra, the third step is a consequence of the Taylor expansion of $\exp(\cdot)$, assuming $\epsilon$ is sufficiently small and the fourth step is the result of the definition of $Z_A$ and involves the consolidation of addition, introducing the common term $Z_A$.

Then

$$q_{\epsilon,i} = \frac{\exp(B_\epsilon x)_i}{Z_B}$$

$$= \frac{\exp(Ax)_i \exp(\epsilon Mx)_i}{Z_A (1 + \epsilon \langle p, Mx \rangle + O(\epsilon^2))}$$

$$= p_i (1 + \epsilon((Mx)_i - \langle p, Mx \rangle) + O(\epsilon^2)). \tag{4}$$

where the initial step is because of the definition of $q_{\epsilon,i}$, the subsequent step is a result of Eq.(3), and the third step is due to the definition of $q_i$ along with the Taylor expansion of $f(x) = 1/(1+x)$ and $\exp(\cdot)$, considering $\epsilon$ as a sufficiently small value.

So, we have that

$$
\begin{aligned}
H^2(P, Q_\epsilon) &= \frac{1}{2} \sum_{i=1}^n (\sqrt{p_i} - \sqrt{q_{\epsilon,i}})^2 \\
&= \frac{1}{2} \sum_{i=1}^n p_i(\epsilon^2((Mx)_i - \langle p, Mx \rangle)^2 + O(\epsilon^3)) \\
&= \frac{1}{2} \epsilon^2 \operatorname{Var}_P(Mx) + O(\epsilon^3) \\
&= \frac{1}{2} \epsilon^2 \nu + O(\epsilon^3).
\end{aligned}
$$

where the first step is the result of Definition 2.2, the second step is because of Eq.(4), the third step the result of definition of $\operatorname{Var}_P(Mx)$ (See Definition 2.3) and the forth step follows from the expression $\nu = \sup_{x:\|x\|_2 \le E} \operatorname{Var}_{\texttt{SoftMax}_A(x)}(Mx)$.

Applying Theorem 3.1 we finish the proof. $\qquad \square$

### A.4 LOCAL UPPER BOUND FOR LEVERAGE SCORE MODEL

*Proof of Theorem 4.3.* We take an $s$ satisfying $c \le s_i^2 \le C$ and $\forall i \in [n]$ that maximizes $\sup_s \operatorname{Var}_{\texttt{Leverage}_A(s)}(w_s)$ and repeatedly query $s$. We need to show that

$$
H^2(\texttt{Leverage}_A(s), \texttt{Leverage}_{B_\epsilon}(s)) = \Omega(\epsilon^2 \nu).
$$

Let $P = \texttt{Leverage}_A(s) = (p_1, \ldots, p_n)$, $Q_\epsilon = \texttt{Leverage}_{B_\epsilon}(x) = (q_{\epsilon,1}, \ldots, q_{\epsilon,n})$. We can compute that

$$
\frac{d}{d\epsilon} q_{\epsilon,i} = (2(I - A_s(A_s^\top A_s)^{-1} A_s^\top)(M_s(A_s^\top A_s)^{-1} A_s^\top))_{i,i}.
$$

Define $W = (I - A_s(A_s^\top A_s)^{-1} A_s^\top)(M_s(A_s^\top A_s)^{-1} A_s^\top)$. Then

$$
q_{\epsilon,i} = p_i + 2W_{i,i}\epsilon + O(\epsilon^2).
$$

Computing $H^2(P, Q_\epsilon)$ we get

$$
\begin{aligned}
H^2(P, Q_\epsilon) &= \frac{1}{2} \sum_{i \in [n]} (\sqrt{q_{\epsilon,i}} - \sqrt{p_i})^2 \\
&= \sum_{i \in [n]} p_i \left( \frac{W_{i,i}}{p_i} \epsilon + O(\epsilon^2) \right)^2 \\
&= \sum_{i \in [n]} \frac{W_{i,i}\epsilon^2}{p_i} + O(\epsilon^3) \\
&= \epsilon^2 \nu + O(\epsilon^3).
\end{aligned}
$$

$\qquad \square$

## B MORE RELATED WORK

**Softmax Computation and Regression**   Softmax computation, a crucial element in attention computation (Vaswani et al., 2017), plays a pivotal role in the development of LLMs. Several studies Alman & Song (2023); Brand et al. (2023); Liu et al. (2023c); Deng et al. (2023c) delve into the efficiency of softmax computation. To improve computational efficiency, Alman & Song (2023) presents a quicker attention computation algorithm utilizing implicit matrices. Similarly, Brand et al. (2023) utilizes lazy updates to speed up dynamic computation, while Deng et al. (2023c) employs a randomized algorithm for similar efficiency gains. Conversely, Liu et al. (2023c) utilizes an approximate Newton method that operates in nearly linear time. Gao et al. (2023) centers on the convergence of overparameterized two-layer networks with exponential activation functions, whereas Deng et al. (2023b); Liu et al. (2023c) explore regression analysis within the framework of attention computation. All of these studies specifically focus on softmax-based regression problems.

