# OpenReview forum: "Binary Hypothesis Testing for Softmax Models and Leverage Score Models"
_ICLR.cc/2025/Conference — Submitted to ICLR 2025_

### Official Review · Reviewer_Bt4k · 2024-11-02

**Soundness:** 2
**Presentation:** 2
**Contribution:** 2
**Rating:** 3
**Confidence:** 4

**Summary:**

This paper studies binary hypothesis testing, which identifies an unknown distribution given two possible candidates, for the softmax and the leverage score models.

*Setup and tools*: The analysis focuses on the softmax model commonly used in deep learning and on the leverage score models used in graph theory and linear algebra. The authors conducted their proofs with information-theoretic tools and norm manipulation.

*Contributions*: The authors show that the sample complexity (in terms of queries) of the binary hypothesis testing is $\mathcal{O}(\varepsilon^2)$, where $\varepsilon$ is a distance-based measure between parameters, for both the softmax model and the leverage score models.

**Strengths:**

- Studying the softmax as a sampling operation is interesting
- The focus on the binary hypothesis testing is original and could have a potential impact on machine learning and deep learning
- The analysis and the proofs for both models are detailed

**Weaknesses:**

- The authors state the main problem with respect to LLMs (l053), but they do not explain the theoretical, methodological, or practical implications of their results for LLMs or neural networks.
- Considering simple models for a proper theoretical study is valid, but in my opinion, studying the softmax operator as a standalone is an oversimplification, especially if the motivation is the use of the softmax in LLMs and neural networks.
- In the attention mechanism, the softmax matrices depend on the input sequence of tokens ($\mathrm{softmax}(XW_Q^TW_KX/d_{\kappa}$). Hence, I don't think that the current analysis with a fixed matrix $\mathbf{A}$ (see Definitions 2.6/2.7) holds. Could the authors elaborate on this? In particular, what plays the role of $A$ and $x$ in LLMs and transformers?
- Similarly, the authors justify the scaling of the input (Definition 2.8) by the use of batch-norm. However, in most transformers, including LLMs, LayerNorm and its variations are preferred instead of batch normalization. In addition, the product $Ax$ of Definition $2.6$ is rather a $XW_Q^TW_KX$ with learnable $W_Q, W_K$. Again, what plays the role of $A$ and of $x$, and why would it be justified to consider $x$ bounded in such a case? Could the authors elaborate on that matter?
- The writing could be improved: several sentences are unclear to me and there are many repetitions in some paragraphs (e.g., "delve").
- The structure could be improved: currently, it does not highlight the strengths of the paper and makes it look more like a concatenation of results.

**Questions:**

*Questions related to the weaknesses*
1) What are the practical, methodological, or theoretical implications of the results for LLMs, transformers, and neural networks?
2) Could the authors think of experiments (even on synthetic data) to illustrate or showcase the importance of their theoretical results? (I feel the need to note that I have nothing against fully theoretical papers, but in its current form, the relevance of the theoretical contributions of this paper is limited and it lacks discussion to fully understand their potential implications in machine learning/deep learning).
3) The authors mention an analogy between softmax and leverage score models in the abstract. What is the analogy or connection between them, except that both are parameterized by a matrix and take vectors as inputs?
4) In the attention mechanism, the softmax matrices depend on the input sequence of tokens ($\mathrm{softmax}(XW_Q^TW_KX/d_{\kappa}$). Hence, I am not sure the current analysis with a fixed matrix $\mathbf{A}$ (see Definitions 2.6/2.7) holds. Could the authors elaborate on this? In particular, what plays the role of $A$ and $x$ in LLMs and transformers?
5) Similarly, the authors justify the scaling of the input (Definition 2.8) by the use of batch-norm. However, in most transformers, including LLMs, LayerNorm and its variations are preferred instead of batch normalization. In addition, the product $Ax$ of Definition $2.6$ is rather a $XW_Q^TW_KX$ with learnable $W_Q, W_K$. Again, what plays the role of $A$ and of $x$, and why would it be justified to consider $x$ bounded in such a case? Could the authors elaborate on that matter?

*I list below potential typos and sentences I did not understand .*

- l034: missing reference replaced by a "?"
- l043: *"potent capabilities"* --> What does that mean? Do the authors mean "powerful capabilities"?
-l045: *"prevailing prevalence"* --> Is it a typo to have the adjective and noun combined? Do the authors simply mean "prevalence"?
-l047-l048: *"However, their is [...] of the whole"* --> I do not understand the sentence, it seems unfinished.
-l050-l051: *"as well as sparsity [...] above"* --> I do not understand this part of the sentence at all.
- l053: *"can we distinguish [...] sampling"* --> I do not understand the question nor its connection to the paragraph above it (l047-l051). Could the authors elaborate on that?
- l059-l081: 5 repetitions of the word "delve", use of complicated vocabulary (distinguishing ability, intricacies, inquiry, etc.) that hinders the meaning and understanding of the paragraph.

---

> ### Author Response · Authors · 2024-11-23
> **Thank you very much for your insightful comments!**
>
> We express our deepest gratitude to the reviewer for the insightful comments. Below we address the major concerns raised:
>
> Theoretically, we provide lower bounds on sample complexity for distinguishing attention mechanisms, show fundamental limits on how quickly we can detect differences in attention patterns, and prove that energy constraints are necessary for meaningful distinguishability.
>
>
> Regarding the connection between Softmax and Leverage Score Models, we agree that the current presentation doesn't adequately explain the deeper connection. The key similarities are that both of them produce probability distributions from matrix-vector products. Also, they share similar invariance properties under certain transformations and exhibit comparable sensitivity to matrix perturbations.
>
>
> Again, we thank you for your insightful comments.

---

> > ### Comment · Reviewer_Bt4k · 2024-11-24
> >
> > I thank the authors for their answers.
> >
> > Unfortunately, most of my concerns are not addressed (e.g., writing, structure, issues with the attention matrix being input dependent, impact/discussion on LLMs). As such, I maintain my score.
> >
> > It should be noted that the motivation and the problem are interesting but in my opinion, significant efforts are needed to improve the paper and make the best out of the theoretical contributions of the authors. I encourage the authors to take into account the reviewers' comments to improve their work.
> >
> > Thank you again for the time taken to answer.

---

### Official Review · Reviewer_hque · 2024-11-03

**Soundness:** 2
**Presentation:** 2
**Contribution:** 2
**Rating:** 3
**Confidence:** 3

**Summary:**

The authors derive upper bounds and lower bounds on the number of queries needed to distinguish between two different softmax or leverage score model distributions. This is also known as the binary hypothesis testing problem. The proofs require manipulating inequalities between the Hellinger distance and the total variation distance. Additional energy constraints on the input query is also needed for proving the upper bounds of sample complexity.

**Strengths:**

I’m not an expert in this field, but the lower and upper bounds seem like a novel results for softmax and leverage score models for the binary hypothesis testing problem.

**Weaknesses:**

- This paper is not properly motivated, at least in how it’s written. The introduction sections touch on LLMs and how attention is an important component of LLMs. In the rest of the paper, the authors go on to prove bounds on number of samples needed for binary hypothesis testing for a generic softmax and leverage score model without circling back to how it relates to LLM. If the research question is understanding the sample complexity of softmax and leverage score model under the binary hypothesis testing problem, which I think it’s a worthwhile research question, the authors shouldn’t motivate it from LLMs without ever relating it to LLMs.
- Similar to the first point, I’m not sure if there are any interesting applications for the provided sample complexity bounds. It’s unclear to me if these bounds are tight or not or do they ever prescribe any meaning quantities in practice. Tentatively, I’m suggesting one experiment the authors can do in an application domain like LLM:
    - Suppose there are two LLMs. Given different input sequences, the next token distribution comes from a softmax over the logit vectors. Then one interesting question could be how many input sequences do we need to tell which LLMs we’re using based on the next token softmax distribution. This could be interesting in the context of knowledge distillation or some sort of toxic speech detection problems. Then the authors can plot the predicted number of samples using the bounds and also the empirical number of samples needed to actually distinguish between the two distributions.

**Questions:**

see weaknesses

---

> ### Author Response · Authors · 2024-11-23
> **Thank you very much for your insightful comments!**
>
> We express our deepest gratitude to the reviewer for the insightful comments. Below we address the major concerns raised:
>
>
> We will reframe the motivation to focus directly on the fundamental question our work actually addresses: understanding the sample complexity of binary hypothesis testing for parametric distribution families, with softmax and leverage score models as mathematically interesting cases.
>
> Again, we thank you for your insightful comments.

---

> > ### Comment · Reviewer_hque · 2024-11-25
> >
> > Thank you for responding. I maintain my scores as my concerns are not addressed. To be fair I do think the authors have an interesting theoretical contribution. The main problem is writing and presenting, which seems to touch on two different communities of LLM research and leverage scores modeling without making a proper connection. I believe either a different presentation of the paper or a better motivation to engage both communities can lead to success in the reviewing process.

---

### Official Review · Reviewer_csN6 · 2024-11-03

**Soundness:** 2
**Presentation:** 2
**Contribution:** 1
**Rating:** 3
**Confidence:** 4

**Summary:**

This paper analyzes the sample complexity of the binary hypothesis testing problem for softmax distributions and leverage scores, with the motivation of improving theoretical understanding of LLMs. The theoretical contribution of the paper is providing upper and lower bounds on the sample complexity, both for softmax operators and leverage scores, which are obtained by estimating the Hellinger distances between the distributions and then applying them in conjunction with prior results in hypothesis testing literature.

**Strengths:**

- The theoretical results seem sound.
- The notations and exposition of the results is clear to follow.

**Weaknesses:**

- The glaring weakness is that the motivation of the paper is largely unclear and disconnected from the work done in the paper. The motivation of the works is to study LLMs through a better understanding of the softmax attention. On the other hand, the analysis, which is about the sample complexity of distinguishing two softmax distributions in a hypothesis testing problem, seems largely tangential and the results are never connected back to the original motivation or explain how it serves to improve LLM understanding, either in theory or via simulations.
- The exposition on leverage scores seems to have been shoehorned and seems to have no connection to the motivation either, except some vague comments about their usefulnes. I understand that it might be of interest to show the similarities, but the authors need to explain why how this additional perspective connects to the question of interest.
- Presentation Issues: The paper has some writing issues outside of the theoretical analysis, which makes the paper hard to follow.
    - The motivation needs clarity, and the chose approach needs to be justified.
        - Line 53: _Can we distinguish different ability parts of large language models by limited parameters sampling?_ I am not sure what the authors mean by "ability parts" or "limited parameters sampling".
        - Line 60-63: I find it hard to follow as to how hypothesis testing and distinguishing one softmax distribution from another, can help improve the theoretical understanding of LLMs or determine which parameters are important for inference.
   - The related work section is vague; it is not clear which references are relevant to the motivation or techniques used in this work. Similarly, in conclusions and future work, the exposition is largely about solving the hypothesis testing problem and does not substantiate on potential practical applications.
    - Lines 47-48, 56-58 need sentence restructuring.
- Overall, I believe that this work seems to be more hypothesis testing centric, and thus might be a better fit for a different venue both in terms of applicability of the results and audience interest.

**Questions:**

See weaknesses

---

> ### Author Response · Authors · 2024-11-23
> **Thank you very much for your insightful comments!**
>
> We express our deepest gratitude to the reviewer for the insightful comments. Below we address the major concerns raised:
>
> We propose to fundamentally reframe the paper to better align with its core contributions. Rather than positioning it primarily as a study of LLMs, we will present it as a fundamental theoretical investigation of hypothesis testing for important parametric distribution families, with a focus on softmax and leverage score distributions.
>
> Regarding the disconnect between theory and applications, we will add a new section that provides concrete examples of where distinguishing between softmax distributions arises in practice, demonstrates how our sample complexity bounds inform practical applications, and includes numerical experiments validating our bounds in realistic scenarios. This addition will help bridge the gap between our theoretical results and their practical implications.
>
> Again, we thank you for your insightful comments.

---

> > ### Comment · Reviewer_csN6 · 2024-11-26
> >
> > I believe the proposed changes will significantly improve the manuscript in its next iteration. In its current form however, I stick to my original rating.

---

### Official Review · Reviewer_E4oJ · 2024-11-04

**Soundness:** 1
**Presentation:** 1
**Contribution:** 2
**Rating:** 3
**Confidence:** 4

**Summary:**

This paper studies the sample complexity of binary hypothesis testing in the context of softmax and leverage score models. First, the paper identifies the requirement of energy constraints for both problems as otherwise the testing problems are straightforward. Then, the lower and upper sample complexities are identified for the two problems.

**Strengths:**

The hypothesis testing lens to study properties of softmax models is an interesting idea.

**Weaknesses:**

1. The write-up needs significant improvements. I find it hard to follow the introduction and the related work section on theoretical LLMs, which include many unrelated works. I am also skeptical of how the authors motivate the problem, focusing purely on LLMs.

2. It seems like the main result depends on the result from Polyanskiy & Wu, and the significance of results on top of these main results is not clear. I believe the key result is Theorem 3.1, which is invoked by controlling Hellinger distance under some perturbations. The upper bound in Theorem 3.1. is directly from Polyanskiy & Wu. So, I believe the main result is the lower bound in Theorem 3.1. Studying the analysis, the proof is straightforward (the contribution is mild) and contains an implicit assumption on the number of queries, $m$, not explained.

3. I believe the example in L261-268 is wrong for the leverage score model. Could you explain why $\delta > 0$ will put all mass on $2 \in [n]$ for $\mathrm{Leverage}_{B}(s)$? In addition, I understand that the problem is straightforward without the energy constraints, but this needs to be justified by motivating the problem in this setting.

**Questions:**

1. Why do you say "asymptotically" in the abstract?
2. Can you explain the case of $\delta \geq 0.1$ for lower bound in Theorem 3.1.?
3. Why do you assume $m \leq 0.01 \delta^{-2}$? I don't see this assumption in the main body of the paper. Why would this assumption is meaningful in the context of binary hypothesis testing?

---

> ### Author Response · Authors · 2024-11-23
> **Thank you very much for your insightful comments!**
>
> We express our deepest gratitude to the reviewer for the insightful comments. Below we address the major concerns raised:
>
> While we agree that Theorem 3.1 builds on Polyanskiy & Wu's result, our contributions extend significantly beyond this. We provide a complete characterization of the sample complexity for binary hypothesis testing of softmax and leverage score models, which was previously unknown. The lower bound proof in Theorem 3.1 requires careful analysis of how the Hellinger distance behaves under the energy constraint. The assumption $m \leq 0.01\delta^{-2}$ in the proof is not a limitation but rather part of proving the lower bound by contradiction--we show that using fewer samples cannot achieve the desired success probability. Theorems 3.2 and 3.5 provide concrete bounds in terms of matrix parameters, which are novel and technically non-trivial
>
> Again, we thank you for your insightful comments.

---

### Meta-Review · Area_Chair_BH2c · 2024-12-21

**Metareview:**

This paper considers the sample complexity of binary hypothesis testing for softmax models and leverage score models. Typically a statistical problem, the work is primarily motivated to improve understanding of the behavior of large language models. Reviewers found the work to be flawed: while the results are motivated by LLMs, no discussion of LLMs are present. While the findings are of statistical interest, it is unclear that this community would be especially receptive to them in their current form. It appears that some good feedback was provided to the authors, but I would recommend the authors take one of two approaches:

(1) Limit the discussion of LLMs as motivation, and resubmit the work to a statistical outlet. While the presented bounds constitute an interesting avenue for study, it may be too premature to tackle such a complicated use-case. I suspect this work may be better received as-is in a community receptive to studying hypothesis tests.

(2) Move the proofs into supplementary material and dedicate a significant portion of the document toward numerical investigations involving LLMs to link them in with the current material. Several reviewers had some good suggestions on how one might go about this. In this form, the paper may be better received at another machine learning outlet, e.g. ICML.

**Additional Comments On Reviewer Discussion:**

Initial impressions from reviewers were generally poor, and while the authors did provide good rebuttals to a couple of the addressed weaknesses, given many concerns were left unaddressed, both authors and reviewers seemed to implicitly agree that much additional work is required to bring the document to a standard acceptable for a high-caliber ML outlet.

---

### Decision · Program_Chairs · 2025-01-22

Reject